# Social and Emotional Learning during Pandemic-Related Remote and Hybrid Instruction: Teacher Strategies in Response to Trauma

Rebecca S. Levine [1,*] , Rebecca J. Lim [2] and Amy Vatne Bintliff [1]

[1] Department of Education Studies, University of California, San Diego, CA 92093, USA
[2] Department of Psychology, The Pennsylvania State University, State College, PA 16801, USA
* Correspondence: relevine@ucsd.edu

**Abstract:** Schools play an important role in fostering student intrapersonal and interpersonal skills and development, also known as social and emotional learning (SEL). This study examined how K–12 teachers used student SEL strategies in remote and hybrid classroom environments during the COVID-19 pandemic, a time of heightened distress and trauma. Survey data were collected from 26 teachers in Southern California and follow-up semi-structured interviews were conducted with 16 teachers. Responses were analyzed from an integrated SEL- and trauma-informed perspective. Themes that emerged included focusing on relationships; building routines and predictability; creating space to identify and share feelings; incorporating movement, mindfulness, and play; implementing culturally affirming practices; providing student choice and leadership; and engaging and collaborating with families. Various challenges associated with implementing SEL during COVID-19 are discussed, including teacher burnout, being unsure who was listening in on class conversations, and feeling disconnected in an online environment. Recommendations for practice and further research are provided.

**Keywords:** social and emotional learning; trauma; COVID-19; remote instruction





## 1. Introduction

In recent years, youth have collectively experienced or witnessed a multitude of traumatic events, including but not limited to the COVID-19 pandemic, racism, and system failures such as police brutality and health disparities [1,2]. The challenges associated with these events take a toll on physical, social, and emotional well-being [3,4]. As such, concerns for child well-being grew at a rapid pace, especially after the pandemic forced school closures in 2020 and instruction moved online. Social isolation, loneliness, losing loved ones, and family stress contributed to significant child anxiety and depressive symptoms [5–7]. Meanwhile, the pandemic magnified existing inequities that disproportionately harmed Black, Latino, and Native American communities [8].

The public education sector has been particularly impacted. Remote learning slowed learning gains, particularly among students from low-income backgrounds [9,10], and increased teacher stress and job ambiguity [11–13]. Teachers witnessed exacerbating inequities and the impact of trauma on student well-being during pandemic-related remote instruction, and they felt overwhelmed and unsure of how to provide support [14,15]. As communities grapple with heightened stressors, there is an urgent call for schools to implement trauma-informed social and emotional learning (SEL) strategies to promote connection and well-being [16–18].

In this study, we investigate the strategies that teachers employed to support student SEL during pandemic-related remote and hybrid instruction, as well as the barriers they faced. The use of in-depth interviews adds teacher voice to the burgeoning quantitative, survey-based evaluations of teachers' experiences during the pandemic [19–21]. This study

was conducted with teachers primarily in under-resourced schools in California, a state with the third-highest student-to-counselor ratio in the US (622 students per 1 counselor), putting many classroom teachers in positions in which they supported student health and well-being in innovative ways [22].

### 1.1. What Is Social and Emotional Learning?

The last two decades have seen a surge in attention toward advancing student SEL at school. SEL is broadly defined as the process of acquiring the skills necessary to develop healthy identities, manage emotions, pursue goals, demonstrate empathy in relationships, and make constructive decisions [23]. Thousands of SEL programs, which include evidence-based curricula designed to promote student SEL, are being implemented in PreK–12 schools across the US, and there is some level of adoption of SEL preparation into teacher education programs in every state [24]. With high-quality implementation, the outcomes of SEL programs include improved social and emotional skills, reduced mental and emotional distress, higher academic achievement, and better overall classroom climate [25,26], although a recent systematic review calls for better reporting standards to be able to meaningfully assess the impact of SEL on subgroups of students, such as those with disabilities and those with minoritized racial identities [27].

In addition to evidence-based programs and curricula, researchers and practitioners have also conceptualized SEL as strategies or pedagogies. As opposed to delivering a packaged SEL program, which can be difficult to implement and may not meet students' needs, a flexible strategy-based approach to SEL may be recommended, in which teachers are encouraged to implement and adapt developmentally appropriate strategies to align with their students' needs and experiences [28]. Other researchers have presented SEL as a type of relational or emotional pedagogy in which consideration of how the classroom functions or how lessons are implemented, and particularly how they promote or inhibit relationships and emotional safety in the classroom, are at the forefront [29,30]. Our study investigates SEL strategies and pedagogies, as opposed to specific SEL programs, for two reasons: (1) immense transition and uncertainty during COVID-19 caused challenges with implementing programming with fidelity, especially in resource-limited settings, and (2) no singular SEL program was used across our participants because we surveyed and interviewed teachers from various schools and school districts, teaching various grade levels, to obtain a broader picture of teachers' SEL experiences.

### 1.2. SEL during COVID-19

By 25 March 2020, all US public schools had closed to mitigate the spread of COVID-19 [31]. At the start of the 2020–2021 school year, 74% of the 100 largest school districts were entirely remote, and by October, the majority of school districts were in hybrid mode, with a mix of in-person and remote instruction. Mounting evidence sheds light on the challenges with remote and hybrid learning, including inequitable access to Wi-Fi [32] and teacher distress and uncertainty [33], as student well-being concerns increased.

A handful of studies explored the landscape of SEL during COVID-19. In a study of 23 novice teachers, researchers found that teachers felt a strong need to focus on SEL and relationships during this time and to find ways to increase student motivation and engagement [34]. Findings from an efficacy study of a teacher-delivered SEL program adapted for online delivery showed promising results, including significant gains in students' social and emotional skills [35]. Another study found that teachers of students with learning differences found creative ways to adapt the SEL program RULER to remote instruction, such as mailing resources home, and implemented SEL strategies like check-ins and sharing emotions in both their classrooms and home lives [36]. Importantly, in order to help teachers feel comfortable and emotionally prepared to implement SEL during COVID-19, support from their school, district [37,38], and colleagues [39] was necessary but not always present. Despite the calls from policy-makers, researchers, and practitioners to improve our understanding and implementation of SEL in response to recent traumatic events,

research on SEL during COVID-19, and specifically teacher perspectives on SEL strategies for remote and hybrid instruction, remains limited. Gaps in the research remain regarding how teachers integrated SEL throughout the school day in these settings, unconnected to a specific curriculum or program evaluation study.

*1.3. SEL and Trauma-Informed Practices*

The COVID-19 pandemic prompted researchers to consider the existing overlap between SEL and trauma-informed practices (TIP) and to call for further integration across fields and frameworks [40,41]. TIP are models of care that consider the prevalence of childhood trauma and its impacts on overall development, health and mental health, learning, and lifelong well-being [42,43]. One of the most widely used frameworks for TIP is the four Rs [43]: *realization* and *recognition*, which builds awareness of the prevalence of trauma and its effects, while acknowledging that trauma can impact learning; *response*, which focuses on individual responses to trauma (i.e., how to respond to students in a helpful way); and *retraumatization*, which invites teachers and staff to create a positive climate and culture of support to avoid retraumatization. Similar to SEL, TIP in schools has the potential to significantly improve student well-being, engagement, and academic achievement [44–46].

As Osher et al. [41] explain, in recent decades, the fields of SEL and TIP have had a converging trajectory. Instead of focusing on individual skill-building, both SEL and TIP have an increasingly expanded scope that recognizes the broader conditions for teaching and learning, including the importance of school climate, well-being, and equity. Common principles of both SEL and TIP are safety, relationships, agency and empowerment, and cultural competence [41]. Ramirez et al. [47], in their content analysis of SEL programs, found similar overlapping practices and strategies; their analysis revealed that trauma-informed SEL practices include creating predictable routines, building supportive relationships, developing student agency, supporting student and adult self-regulation, and engaging in individual and community identity development [47]. At the core of both SEL and TIP is an understanding that children's emotional well-being, connectivity, and social relationships play pivotal roles in their decision making, ability to learn, long-term well-being, and overall health [25,48].

An example provided by Osher et al. [41] of an integrated SEL and TIP model is the HEARTS framework: Healthy Environments and Response to Trauma in Schools [49]. The six core principles of the HEARTS framework are: understanding trauma and stress; cultural humility and equity; safety and predictability; compassion and dependability; empowerment and collaboration; and resilience and social and emotional wellness. Our study builds upon research by Osher et al. [41] and Ramirez [47], and frameworks such as the HEARTS model [49], to analyze teachers' use of trauma-informed SEL strategies and learning activities during remote and hybrid instruction. A trauma-informed SEL perspective was selected for analysis due to the emerging research on the impact of COVID-19, a collective trauma, on students [50].

While the convergence of SEL and TIP is promising, we want to call attention to the potential harm that SEL can cause when it is not implemented in a trauma-informed manner. Students experiencing trauma tend to have higher emotional reactivity, more difficulty recognizing emotions, and lower self-concept; in these cases, emotional responsiveness from adults at school is even more important to establish emotional safety and help students "experience themselves as positive, appreciated, and effective members of the community" [51] (p. 39). Scholars have called out another concerning trend of SEL being used to control or police students' behavior, particularly students of color, and the need for SEL to be culturally affirming and equity-focused in order to promote wellness for diverse school communities [52,53]. Thus, it is critical for teachers and administrators to implement SEL within a culture of care, safety, and support, and research is needed that can help continue to move the trauma-informed SEL literature beyond theory and into practice.

Our study contributes to the literature by investigating teacher experiences of SEL during COVID-19, providing evidence of what trauma-informed SEL strategies looked like in remote and hybrid environments. Our study is exploratory in nature in order to capture teachers' lived experiences, including their thoughts, feelings, observations, and behaviors related to SEL during this time. We also investigate barriers to trauma-informed SEL in order to recognize limitations and constraints that teachers face in the pursuit of promoting a safe environment for SEL within their school communities. Our aims are that these findings are ecologically relevant for teachers, administration, and students, and that they contribute to the growing literature on both the conceptualization of trauma-informed SEL and its practical implementation.

## 2. Materials and Methods

### 2.1. Researcher Positionality

The authors of this study comprise three researchers associated with the same public university: a white, female teaching professor who previously worked as a teacher at alternative secondary schools with training in TIP before receiving a PhD in educational psychology; a white, female PhD student who is a licensed social worker with clinical training and experience working with youth who have experienced trauma; and an Asian American, female former undergraduate student and current research coordinator with a background in psychology, education, and direct K–12 academic support. All authors were students and/or teachers during COVID-19 remote and hybrid learning.

### 2.2. Procedures

In May and June 2021, teachers in Southern California were recruited to participate in this study via an email listserv of a university service-learning community-based partnership. All teachers on the listserv (79) were teachers in local K–12 schools representing multiple school types (private, public, public charter) and districts. The recruitment email described the study, provided a consent form, and included a link to complete a Qualtrics survey. The survey asked basic demographic questions and a series of multiple-choice items and open-response questions related to SEL. The survey, taken online at a time and place of the participant's choosing, took approximately 15 min to complete. A total of 26 out of the 79 invited teachers completed the Qualtrics survey—a response rate of 33%. This response is lower than expected, which may have been due to teachers feeling overwhelmed during the pandemic. We sent two survey reminders via email during the last few weeks of school and, in efforts to alleviate stressors, we conducted interviews during the summer months when teachers were on vacation.

At the end of the survey, teachers were invited to provide their contact information if they were interested in participating in a 30 min Zoom interview to elaborate on their experiences with SEL during remote and hybrid instruction. A total of 21 of the 26 teachers who had completed the survey (64%) expressed interest in a follow-up interview. The research team contacted all teachers who expressed interest, and interviews were successfully scheduled and conducted with 16 teachers during the summer of 2021. (Five teachers did not respond to attempted contact.) The semi-structured interview focused on student well-being and SEL during the 2020–2021 school year. Interviewees were compensated with a $25 gift card for their participation. All research procedures were approved by the university's Institutional Review Board.

### 2.3. Survey

The motivation behind our survey items was to collect demographic information and establish teachers' existing familiarity and beliefs around SEL. The survey included multiple choice and open-ended questions developed by the research team and informed by prior SEL research. The first set of multiple-choice questions asked for participant demographic information, such as gender, race, and ethnicity, as well as professional teaching background and information about their current employment setting. The development of the next set of

multiple-choice questions was grounded in previous work on teachers and SEL [54,55] and followed a similar procedure to that employed by Buchanan et al. [54] in their pilot survey of teachers' knowledge, perceptions, and practices of SEL. In this section, participants were asked multiple-choice questions such as, "How familiar are you with the concept of social and emotional learning (SEL)?" and "Where have you received information about SEL?" Participants were also asked to indicate, on average, for each month of the 2020–2021 school year, whether they had been teaching remotely, in a hybrid format, or in person.

Open-response items were also included in the survey because of the exploratory nature of the study and to reduce bias in responses due to pre-determined response options. Open-response items included: "How has remote learning impacted your ability to engage in social and emotional learning with your students?" and "What strategies, if any, have been helpful in supporting your students' social and emotional learning or well-being during the pandemic?"

*2.4. Interview*

The semi-structured interview questions, while similar in nature to the survey, allowed us to probe more deeply to understand teachers' SEL experiences and strategies used during remote and hybrid instruction. The interview questions were developed by the research team based on the feedback that university faculty members were receiving from student teachers, graduates, and teachers who partnered with our university regarding remote learning. We were hearing about the difficulties associated with remote instruction and began to wonder how teachers were thinking about and implementing SEL in this unprecedented moment in education. Example interview questions include: "Please describe your familiarity with social and emotional learning, or SEL"; "Please describe your thoughts about SEL during remote instruction"; and "What pedagogy, strategies, or tools were effective during remote instruction to meet the diverse non-academic needs of your students?" Interview questions probed for both remote and hybrid situations.

*2.5. Data Analysis*

We ran descriptive statistics on the multiple-choice survey data using SPSS version 28. This enabled us to summarize various aspects of our data and provide details about our participants [56]. In particular, we ran frequency analysis on each variable of interest. Our quantitative data, therefore, helped to establish participant characteristics that were relevant to this study, serving to contextualize and supplement their qualitative responses.

Open-response survey items were treated as qualitative data along with the interviews. Qualitative data were analyzed using MAXQDA. Interviews were transcribed, all identifying information was removed, and pseudonyms were used in place of participant names. The research team read each interview and coded the data both inductively and deductively, a strategy referred to as abductive or complementary coding [57]. Using both inductive and deductive analyses can "help the researcher focus on the research purpose, as well as paradigmatic, theoretical, and conceptual lenses" [58], p. 146. First, we conducted attribute coding of the data. Attribute coding is a process typically undertaken in the first round of coding that captures essential information about the data and the participants, which is helpful for data management and future reference. In our study, attribute coding involved coding the data based on their source (interview or survey) and any relevant demographic data that were shared by the participants qualitatively. Second, we deductively sorted data into categories that supported our initial research questions through a process of reading and re-reading the data alongside the trauma-informed SEL literature [59]. The broad categories were (1) trauma-informed SEL strategies and (2) challenges with implementation. We then analyzed the data inductively using a combination of in vivo codes stemming from the words of the speaker [60] and open coding. We discussed the inductive and deductive codes, organized them into clusters, and created heading titles that became our themes (see Appendix A). We met as a research group to interrogate the themes, collapse them, and rename them when needed, via the back-and-forth movement between our research

questions, the existing literature, and the participants' words. As we moved towards our interpretation, we wrote definitions for our themes and counted the phenomena occurring in the data [61]. Regular team meetings and reflection memos, written after each interview and throughout the analysis process, helped us to examine our positionality in relation to the work, the ways in which we were making meaning of the data, and our pre-existing biases, enabling us to consistently return to the data and teachers' voices and experiences.

*2.6. Participant Characteristics*

Of the 26 participants, 22 were female (84.6%), two were male (7.7%), one was non-binary/third gender (3.8%), and one preferred not to say (3.8%). In terms of racial background, 17 participants were white (65.3%), five were Asian (19.2%), one was Black (3.8%), one preferred not to say (3.8%), and two selected "other" without further clarification (7.7%). Eight participants were Latino/a/x (30.8%). More than half of the teachers ($n = 16$, 61.5%) reported that they worked in schools with high levels of poverty (with 75% of students or more receiving free or reduced-price lunch). Additionally, all teachers ($n = 26$, 100.0%) taught in a remote or hybrid environment for at least one month of the school year, while a few teachers ($n = 4$, 15.4%) taught fully in-person at any point during the school year. See Appendix B for demographic information by participant. We also present the descriptive statistics of the demographic information in a table format (Appendix C), organized into two groups: participants who completed the survey only ($n = 10$) and participants who completed both the survey and the interview ($n = 26$).

Teacher SEL Background

The survey data provided evidence that our participants had a high level of awareness of, belief in, and investment in SEL practices. Indeed, all teachers reported using SEL strategies in their classrooms prior to COVID-19 ("often": $n = 20$, 76.9%; "sometimes": $n = 5$, 19.2%; "rarely": $n = 1$, 3.8%). According to the survey results, most teachers (n = 19, 73.1%) reported that they were "very familiar" with SEL, seven (26.9%) reported "a little familiar," and no teachers reported "not at all familiar". When asked about where they had received information about SEL, nearly all teachers ($n = 22$, 84.6%) reported that they had participated in SEL trainings/workshops (e.g., Second Step, Responsive Classroom, Sanford Harmony, credential courses); half ($n = 13$, 50.0%) had learned about SEL in graduate school; nearly half ($n = 12$, 46.2%) had learned about SEL online (excluding social media); and eight teachers (30.8%) had learned about SEL on social media.

In our interviews, teachers elaborated on their SEL background and training. Shelby selected a teaching credential program specifically for its emphasis on SEL, explaining, "I feel like it helped to boost and confirm beliefs that I have that are important around social and emotional learning." Others, like Rachel, experienced a more patchwork introduction to SEL: "Just my own research, and I know what works for me. There's no quote 'curriculum' we have. We've found a lot of things out on TPT [Teachers Pay Teachers] or through our counselor in the past few years." Regardless of their varying levels of previous exposure to or training in SEL, all teachers ($n = 26$, 100.0%) indicated on the survey that they agreed with the statement, "It is part of my role to support children's SEL," and most teachers ($n = 23$, 88.5%) agreed with the statement, "I would like more guidance on SEL strategies I can use," suggesting that the teachers in this sample felt a degree of responsibility for implementing high-quality SEL with their students.

## 3. Results

*3.1. Centering Students' Needs in Curricular Choices*

During the 2020–2021 school year, teachers recognized that families were coping with heightened traumatic events, including job and housing loss, family deaths, difficulties accessing education, and social isolation, especially in low-income communities. They also spoke of recognizing symptoms of trauma within their students such as disengagement, feelings of isolation, symptoms of grief, sleeping during class, and difficulties forming new

friendships or connections. In response to the numerous ongoing challenges and the shift to remote instruction, nearly all participants remained committed to SEL. Kendall explained:

> I kind of made a decision early on . . . the academics, they're going to be there, but they're not going to be my priority. These kids need connection, they need to feel, you know, a part of a community, and they need to feel like we're all in this together. That was really important for me: to make sure that their social–emotional needs were met.

In fact, many teachers expressed that SEL was more important now, during a time of heightened trauma, than ever before. Kendall continued, "The longer it [the pandemic] went, that's when I really started to see the toll that it was taking on students . . . I mean SEL was, like, non-negotiable".

However, the shift from in-person to remote teaching required a new approach. As Shelby wrote on the survey, "The pandemic has forced me to become more creative with implementing SEL strategies, and even redefining what they look like online." The following sections describe the various themes that emerged from the qualitative data regarding teachers' trauma-informed SEL: focusing on relationships; building routines and predictability; creating space to identify and share feelings; incorporating movement, mindfulness, and play; implementing culturally affirming practices; providing student choice and leadership opportunities; and engaging and collaborating with families.

### 3.1.1. Focusing on Relationships

Relationships were of the utmost importance to the teachers in our sample. During remote instruction, many teachers implemented creative strategies to build relationships with each and every student, including sending students personal notes and voice recordings, holding one-on-one Zoom meetings (especially "after returning from a school break to check in about mental well-being", shared Shelby on the survey), and stopping by students' homes on occasion. Teachers also paid attention to smaller details that they suspected would help students feel more connected, like spending extra time providing personalized feedback on assignments, or making sure to respond to every student who engaged in online chat features. Getting to know students enabled teachers to respond to in-the-moment SEL needs, even in a remote setting. As Sandra explained:

> I could just, you know, take a breath and say, 'Okay, we don't need to keep talking about the quadratic equation. We're going to do mindfulness'. Over Zoom I could just tell, you know, the responses were dwindling, less chats coming in, so like, 'Okay everyone, just stand up, stretch . . . '

Teachers also made a concerted effort to foster peer-to-peer relationships, recognizing that many students were more disconnected from their peers than ever before. Breakout rooms helped create "a friendlier space than seeing, you know, 26 little squares". However, without any structure, teachers reported more silence, awkwardness, and surface-level connection. To help reduce these barriers, teachers used conversation starters, show-and-tell, and scavenger hunts, along with clear instructions and expectations.

Fostering peer-to-peer connection in a hybrid format was challenging. Teachers came up with strategies to encourage students at home to engage with their in-class classmates: "I took a pair of headphones and I put a kid over here and I was like, 'Okay, you guys are going to play your math game together on your screen . . . And I just sanitize the headphones, and tomorrow you'll have a new guy,'" said Rachel, in attempt to help the students at home meet everyone in the class.

### 3.1.2. Building Routines and Predictability

Teachers recognized that building routines and predictability was critical during times of heightened stress, uncertainty, and unpredictability. Routines, including having meetings every morning, reviewing the day's schedule with students, and classroom charters and norms, helped students to understand expectations and feel more prepared to engage

both interpersonally and academically. Three elementary school teachers—Priya, Erin, and Shelby—referenced prior training in Responsive Classroom (RC), an evidence-based SEL approach. They described how elements of their RC morning meeting practice [62] transferred online, including greeting each other by name, sharing about important life events, a group activity for SEL skill development and group cohesion, and a morning message to prepare students for the day ahead. As she reflected on her students' home lives during COVID-19, Erin explained why she chose to continue implementing daily morning meetings:

> I understand they don't have to have, like, as structured of a routine, in terms of, like, packing a backpack, and eating breakfast . . . a lot of them would just sort of roll out of bed and open up their device . . . so I would do a morning meeting, like the full-structured—it has like four parts that takes about 30 min to complete. I normally don't do that in the classroom, but I just found we needed something to, like, get the kids engaged and accountable for showing up on time and actually speaking and engaging with each other.

Whole-group conversations were also a time to develop and reinforce classroom norms. Frederick created a classroom charter, similar to what he would have done in person with his class; and Shelby described how she made minor adaptations to the remote setting: "We had our whole group norms kind of throughout the year but would also develop small group norms every time we shifted our small groups. We'd come up with small group names and colors and icons and things to really, kind of, feel like our own." Additionally, Sandra, who taught high school, found that using an online learning platform helped teachers and students alike to stay organized and improve their self-management skills, as it helped provide a sense of structure: "They were able to see the Google Classroom, to see what they're missing, what they turned in, so that was a lot nicer and I think we got into a good groove." Routines and structure provided consistency and helped students feel more in control during a time of immense change.

### 3.1.3. Creating Space to Identify and Share Feelings

Providing time for students to identify their emotions, reflect, and share took a number of forms during remote and hybrid learning. Tools included SEL journal prompts, mood/feelings charts, a feelings survey accessible at any time on the class website, one-on-one conversations, and whole group meetings. Reading and reflecting on characters' emotions in books or in videos was another strategy that translated well from in-person to remote environments. Morning meeting was a popular time for discussing emotions. As Anika explained on the survey, "I continued morning meeting, on Zoom and in person, so that each child had a chance to share with the class how they were feeling and every child was greeted and heard by the class." Importantly, teachers used these various feelings check-ins as a tool to deepen connections. Kendall explained:

> Every single morning, before I even started the day and going over our norms, it was like, 'How are you doing today?' And the kids would show me on their fingers like 1, 2, 3, 4, 5. And each one was like a different emotion, so if I saw a 5 like they were upset or sad about something, I would jot that down. And then that would be a child that I touched base with later on in my day. 'Hey I noticed you were at a 5 earlier today, did you want to talk about that?', you know, and that's when those connections would start to build, like, 'Yeah my grandma passed away, she had COVID' . . . I was able to have just those intimate connections and conversations that were really raw and meaningful to them and really gave me insight into what they were going through, which would then help structure the SEL for the next days.

With this new knowledge about feelings and important events in their students' lives, teachers were able to make adaptations and accommodations to meet their students' needs, just as they would have done in person. Their check-ins opened the door for

conversations and support around grief, isolation, and other effects of the pandemic. In a remote environment, teachers found that having intentional practices to discuss feelings was paramount, as they no longer had the casual opportunities for side conversations or visual emotional cues that are available in person.

An additional strategy commonly reported by teachers was modeling. Teachers regularly modeled emotional self-expression and regulation, especially on Zoom, which was often an exhausting or frustrating experience for all involved due to connectivity or other technology challenges. Rachel recalled, " . . . And myself, what's that term? Metacognition. 'Gosh, you guys, I'm, like, really stressed out right now. I think I need a break.'" By expressing their own emotions in these moments, teachers created a safe space where a range of emotions was welcomed. Teachers also adapted old lessons to new environments, such as Anika, who shared on the survey, "I taught students about how to recognize emotions even behind a mask, which was a fascinating adaptation of our regular emotion–recognition lessons".

### 3.1.4. Incorporating Movement, Mindfulness, and Play

Teachers incorporated movement, mindfulness, and play as a means of taking breaks from the physically sedentary and mentally exhausting aspects of sitting in front of the computer for school. Many spoke to the ease of implementation and breadth of available resources online; teachers could easily look up a diverse array of engaging movement- or mindfulness-based activities and share them with the class in real time. As Ashley said on the survey, "Without the pandemic I wouldn't have thought to find YouTube videos with breathing exercises!" Some teachers implemented mindfulness with fidelity, while others used it to respond to needs on a given day (e.g., if students were distracted, frustrated, or exhausted with Zoom). Kendall said:

> It was powerful seeing the kids actually on their screens, closing their eyes. You could see almost, like, the physical reaction too, of them relaxing . . . I knew they're going through trauma right now, they're losing family members left and right, and this is going to impact them tremendously. So we did the breathing exercises and then we would reflect . . . 'How did you feel today? How did this exercise help you? Is this something you would try again in the future?'

However, Ashley, a teacher of Deaf students, highlighted an issue regarding online videos: "Not all of them are very Deaf-friendly. Some of them are very verbal and, like, directions all spoken. So we would go through, and we made a Google Slides deck of Deaf-friendly activities that were really, like, more dancing, and easy for the kids to follow along." This highlights an equity issue related to availability and accessibility of resources in an online environment.

Last, games were of utmost importance. Not only did they provide a needed break, but they also contributed to joy and connection. Loren said, "[We did] lots of play-time, I'd say, like, building community values. We'd play so much Among Us . . . I had parents emailing me like, 'That's the only time I've seen my kid have fun all week, so thanks for hosting that.'" Teachers described art activities, music, dancing, and virtual field trips that helped bring joy to the students' and teachers' days. During this time of collective trauma, teachers tried to restore moments of joy through shared playfulness and creativity.

### 3.1.5. Implementing Culturally Affirming Practices

A few teachers also spoke to the importance of implementing equity-focused and culturally affirming practices in their classrooms, especially amidst ongoing social issues, system failures, and racial trauma. The online environment presented an opportunity for students to connect with other students around the world via global student networks, where they were "talking with people from around the world, making connections, what racial and social justice looks like," described David, an elementary school teacher. Other teachers who already worked at schools with strong social justice values were able to maintain that focus during remote instruction. Shelby explained her school's philosophy: "We're

wanting to develop a sense of pride in who we are, individually and as a collective . . . It is a journey, but I think that that belief is carried through how we're designing projects, designing our work, and designing those conversations." One example of culturally responsive SEL was offered by Morgan, when she asked her students:

> 'Hey, where does your family come from? What do you think your ethnic background is?' . . . 'What city in Mexico is your family from?' so that I could try to find, like, somebody from that city even because I just think it's really important that they believe that they can do it [be successful in school]. I think the most important part of my job, actually, is to make them believe that they can do it.

### 3.1.6. Providing Student Choice and Leadership Opportunities

Teachers described various examples of providing student choice and leadership opportunities. When possible, students were offered choices regarding their schedule ("We did lots of voting on what to do first or wait until the next day") or their work environment ("Do you want to work alone, do you want to work with a partner, do you want to work in a group?"). In another example, Rachel realized that, instead of assigning the Fun Friday breakout rooms herself, "Eventually we just said, let participants choose where to go, because it literally takes me like five minutes to be like, 'Where are you going, where are you going?' And then they just had so much more fun popping around, and they just loved it". Teachers described how providing choices and incorporating students' preferences helped to improve self-awareness, autonomy, and sense of belonging, and at least for one student in Sarita's class, improved attendance: "I had them do a survey of what their favorite song was. So then in the mornings, I'd have like a huge playlist and I'd put their different songs on. One of the kids told me, 'The reason I come in on time is because I want to see if you play my song.'"

Leadership opportunities included implementing classroom jobs adapted for remote learning, such as calling on hands or being chat message readers. Sonjia's classroom had a "Boss of the Day": "They would tell me what to, you know, tell the date, tell the weather, and then this, and all that little stuff that you normally do in an elementary school because I felt like it provided some normalcy." Meanwhile, Elena entrusted one student with co-host responsibilities each day: "They would have like power over the meeting, and then they would like let people in, and they were responsible for dropping things in the chat."

### 3.1.7. Engaging and Collaborating with Families

Many teachers found that they were more connected to families than they had been in previous years. Teachers were in touch with parents/guardians more regularly to solve technology issues, check in on how students were doing, and, ultimately, partner with them to support students' academic and social and emotional needs. As Sandra explained, "Since everyone was at home, it was so much easier to contact parents. I could just send them a quick text." Multiple teachers described how hosting parent—teacher conferences via Zoom allowed for increased accessibility and flexibility compared to previous years, as parents/guardians did not need to take time off from work to travel to and from the school. Perhaps most importantly, teachers found that, especially during this time of uncertainty and worry, taking time to genuinely and purposefully connect with families fostered stronger systems of support that lasted throughout the year. Kendall and her partner teacher set up individual meetings at the start of the school year:

> We took the 10 min to talk to these families, get their background, like, how they were feeling about this whole situation because it was new to everybody. And just giving them that reassurance that their kids were in good hands, and we were going to make the most of it . . . That really did just set the year off on such a positive note.

These family meetings provided an opportunity to form relationships, share information about routines and learning goals, and express any concerns about personal or academic issues, enabling teachers, students, and families to feel like a team as they navigated new learning environments and pandemic-related hardships.

*3.2. Barriers to SEL Implementation*

While most teachers recognized the relevance of SEL in the context of the pandemic, they described numerous barriers to implementing SEL strategies in remote learning environments. These challenges included privacy concerns, connectivity challenges, and teacher burnout. As Loren summarized about SEL during COVID-19: "So, some weird mix of positive and negative, but definitely a net loss. Like, I'm ready to do normal teaching." Many teachers echoed this sentiment, expressing that they felt that remote SEL was inferior compared to in-person.

3.2.1. Disconnection

Teachers frequently brought up how teaching remotely hindered their ability to build deep and consistent relationships with their students in virtual learning environments. As Priya explained, "I couldn't feel the kids." This was particularly worrisome for many teachers, who recognized that students, especially the younger ones, were potentially missing out on interpersonal experiences critical for social and emotional development. Many students had Wi-Fi issues that prevented them from fully engaging in an online format, and others had their cameras turned off for extended periods of time.

Teachers explained how being unable to see their students presented a variety of challenges, including an inability to fully engage students, being unsure if students were present, and not having access to visual cues for reading students' emotions. Alexis explained on the survey, "Not being able to view body language impacted my ability at times to engage in social and emotional learning with students." Even when students had their cameras on, teachers often struggled to encourage participation and keep students fully engaged. As Loren reflected, "The kids that really needed it [SEL] the most didn't necessarily get it because you can't force them to log on. You can't force them to do the things over Zoom." Teachers discussed the challenge of facilitating peer-to-peer connections as well, citing their difficulty with figuring out how to overcome the awkwardness and social isolation associated with staring at little boxes on a screen all day. "We didn't have a community," Priya said. All of this felt defeating to teachers. Sarita said, "With online, it was hard for me to keep up with 'Hi good morning, how are you? What am I looking at? Am I looking at an icon? Are you really there?'"

As schools re-opened, hybrid and in-person learning environments during COVID-19 came with their own challenges for relationship building. With masking, physical dividers, and physical distancing requirements, students were often "stuck to one area," which made it difficult to have collaborative group work. Teachers who were used to high-fives, hugs, and circle time on the rug struggled to adjust.

3.2.2. You Didn't Know Who Was Listening to You

In a year so full of emotional challenges, isolation, and injustices, teachers felt that they were limited in their ability to converse candidly and authentically with their students. Some teachers were concerned that they might have been missing opportunities to refer students to counseling. It was almost impossible to tell who might have been listening off-camera, and students living in difficult situations, such as abuse or neglect in the home, could not ask for the help they might have needed. This was problematic for trauma-informed SEL, as students' physical and emotional safety was uncertain. Natalie explained, "You didn't know who was listening to you . . . Maybe some kids couldn't tell us the truth because someone was sitting next to them." Ramona expressed a similar concern on the survey, reflecting that if the teachers needed to have a difficult conversation about an SEL concern with a student, "The children could just walk away or turn off the computer."

On other occasions, parents who had been overhearing classroom practices interfered with SEL implementation: "I had a parent tell me like, 'Oh, like my son's not going to show up for this part [SEL] because it's a waste of time,'" shared Anika. Although some teachers in our sample reported parent support and gratitude for SEL, other teachers in our sample were highly aware that they had an additional audience of parents or other family members, and that student engagement in SEL activities may have been impacted by family members' values and opinions.

3.2.3. Teacher Burnout

Another barrier to SEL implementation was teacher burnout. One teacher described the year as "extremely depleting, soul-sucking, overwhelming, hard," while others described the year as "depressing," and "out of control." On top of the emotional stress of feeling powerless and ineffective at teaching students in remote and hybrid environments, they described the painful physical effects of sitting in front of a computer screen for hours at a time, day after day. In addition, many teachers worked longer hours, providing technology support or adapting lessons to online instruction. In our sample, teachers who worked in low-income communities were particularly worried, frustrated, and sad about inequities that were exacerbated by the pandemic. Rachel shared on the survey that she had socially distanced "pick-up parties" for students to collect materials needed for remote learning, but, as she explained, "if families did not show, that severely impacted what my kids could do."

Some teachers described masking their own frustration and exhaustion in front of their students, hoping to preserve a sense of peace and safety in their virtual classrooms amidst the unprecedented circumstances. As ever-changing demands of online teaching remained, "putting on that front" became a daily routine. On the survey, Naomi reflected, "I'm having a hard time so I have to be patient with myself before I can take care of the kids." A few teachers in our sample admitted that SEL was put on the back burner as a result of their exhaustion. Natalie shared:

> To be honest, we were so overwhelmed just trying to get our lessons up on the website and teaching anything that we possibly could, that, you know, I didn't want to go searching around for, 'What can I do to make them feel better?' you know, because I mean, I just wanted to get the day done and get off the computer.

Larger concerns regarding lack of administrative SEL support also contributed to burnout. Some teachers in our sample felt alienated from pandemic-related goals or decision-making processes (e.g., learning technology tools versus learning TIP or SEL), and teachers reported feeling the burden of caring for students and their families in the day-to-day context, seemingly alone, or with a small group of colleagues. Although some teachers reported strong administrative teams, administrators were busy with scheduling, creating new policies, delivering devices for families, and other tasks that kept teachers feeling fairly isolated. Only one teacher spoke of partnering with a school counselor, and others highlighted the need for more counselors in California who have received training on grief and coping.

## 4. Limitations

Certain limitations must be acknowledged within the present study. Since the majority of the participants were white, female elementary teachers, this study lacks demographic diversity, although these demographics are representative of the current teacher workforce in the US [63]. In addition, this study lacks student voice, as survey and interview data were only collected from teachers. Future research should incorporate students to understand their perceptions of SEL and well-being, especially in relation to trauma. Finally, the "opt-in" study design and the response rate of 33% could indicate a response bias towards teachers who had strong feelings surrounding remote and hybrid SEL. Thus, while these findings cannot be generalized to a larger population of teachers, the study does shed light

on a select group of teachers and their successes and challenges regarding remote and hybrid SEL during COVID-19.

## 5. Discussion

This study reveals both the hardships that teachers and their students experienced during COVID-19 remote and hybrid instruction and the strategies that teachers employed to support their students' SEL. Teachers recognized the need to prioritize SEL by focusing on relationships, identity development, emotions, trust, and safety within their classrooms. From the teachers' perspectives, SEL was incorporated into classroom instruction to address student's needs, as well as their own needs, for community building and social support. Teachers reported using mindfulness techniques to regulate emotions as well as to re-engage students in focused learning. Other strategies created fun and enjoyment in their online classroom or worked to mirror and extend daily activities and classroom routines in order to build feelings of safety.

These strategies align closely with a number of trauma-informed recommendations put forth by Osher et al. [41] and Ramirez et al. [47], as well as the HEARTS framework by Dorado et al. [49]. Regarding the four Rs [43], teachers spoke often about their *realization* that pandemic-related events, such as deaths in the family, parental job loss, and isolation, had the potential to negatively impact students' well-being, learning, attention, and mental health. They would pause and *recognize* students who needed additional support. Teachers also reported varying their *responses* to incidences of student frustration by being more patient, more creative, and more understanding. In terms of limiting *retraumatization*, teachers focused more attention on ensuring that students felt cared for and supported through both teacher–student relationship building, small group relationship building, and whole class engaging activities. They also built in opportunities for student autonomy and choice. This is the first study to our knowledge that sheds light on teachers' descriptions of what these strategies looked like in practice during pandemic-related remote and hybrid learning.

The current study has broad implications that extend beyond the circumstances at the time of data collection. First, we want to emphasize that online teaching and learning in K–12 was steadily growing prior to the pandemic [64], with evidence that this growth will continue; according to a RAND report, about one in five US school districts plan to offer remote learning options post-pandemic [65]. Meanwhile, we know that the effects of a traumatic experience do not resolve once the event is over [66]. Psychologists project that we will continue to see the adverse effects of COVID-19 on well-being for years to come and emphasize the need to respond with support and opportunities for healing, for students and teachers alike [67,68]. While most of our sampled teachers stated that they prioritized SEL during COVID-19, future research is needed to document whether and how SEL prioritization can continue, considering the complex demands of learning loss and high expectations of standardized test improvement [69]. Finally, the findings of this study related to teacher strategies in response to trauma, as well as barriers, are relevant for how to prepare for and respond to current and future collective traumatic events.

We have a number of recommendations based on our study. First, we want to highlight various remote SEL strategies that may also enhance in-person SEL instruction. For example, teachers may consider continuing to offer Zoom meetings to meet the diverse needs of families, use tools like online global networks for cross-national and cross-cultural SEL, and post surveys on a class website to provide students with low-barrier options for communicating with teachers and counselors. In remote or hybrid classrooms, however, a notable finding is that there may be fewer organic opportunities for connection as well as constraints related to not being able to see students' expressions or body language, resulting in disconnection and detachment. Since the core tenets of TIP are attunement (i.e., being aware and responsive to students' needs) and relationships [70]—the opposite of disconnection and detachment—it may be even more critical to intentionally design remote and hybrid trauma-informed SEL opportunities that are dedicated to relationship building, identifying and sharing feelings, and joy.

Next, we want to emphasize the importance of collaboration among teachers and counselors for trauma-informed SEL and elevate the call for more counselors in schools [71]. According to our study, co-facilitation and co-planning among teachers and counselors were rare occurrences. Without support, teachers implemented piecemeal activities found from various sources online, such as YouTube. This is reflective of a larger trend within education as teachers look to social media for lessons of varying quality [72], and indicates that there may be ways to improve the dissemination of education research for uptake in K–12 classrooms. Therefore, while we recognize that teachers should be trusted to respond to the needs of their students through flexibility in their planning, we do not know if the activities they were implementing had been assessed by someone with an SEL or trauma-informed background, or if the teachers were prepared to address any adverse outcomes (e.g., increased social anxiety [73] or traumatic re-experiencing [involuntarily reliving a traumatic event] [74] during or after mindfulness exercises). This is not to say that check-ins or mindfulness should not be used as part of SEL, but instead showcases risks that may be heightened when teachers are not trained, use unvetted materials, or do not have ongoing partnerships with counselors. As our survey shows, nearly all the participants would like more SEL guidance, suggesting an openness for further training and collaboration.

The lack of a cohesive school- or district-wide SEL approach also limits support that teachers can receive from colleagues or administration. Recent research conducted during COVID-19 confirms the importance of school/district guidance, support, and commitment for SEL implementation [37–39]. For example, school staff can work together to develop a proactive strategy for engaging with families, including more transparent and accessible communication and opportunities for school–family SEL partnerships, which may help reduce parental resistance or concerns. Structures and policies at the school/district level can also promote teacher well-being and reduce burnout by removing the burden on teachers to find their own SEL materials, as well as by distributing the responsibility that teachers felt to maintain their students' well-being during collective trauma. Reducing teachers' feelings of burnout is critical for SEL because we know that student SEL starts with how the adults are doing [75], with research showing that teachers' emotions are an important predictor of how students are feeling [76]. Therefore, systemic responses to improve teacher well-being and connection must be at the forefront so that all can thrive, especially after months or years of many teachers feeling distressed, disconnected, and considering leaving the profession due to the challenges of teaching during COVID-19 [77].

Finally, we want to highlight the need for equity-focused SEL, such as transformative SEL [78]. Transformative SEL involves students in labeling power dynamics and the effects of racism and other systematic injustices, and developing solutions with supportive adults, to promote community well-being. The injustices described by our participants, such as poor Wi-Fi connection and housing insecurity, demand a look at the very systems that teachers and students negotiate. In order to promote equity, trauma-informed SEL needs to provide the school community with the tools to interrupt the systemic inequities that are linked to trauma.

## 6. Conclusions

This study sheds light on teacher strategies for promoting SEL during COVID-19, a time of collective trauma and uncertainty, as well as the barriers and challenges that teachers faced related to SEL during this time. By analyzing interview and survey data from a trauma-informed perspective, this article provides new insights into trauma-informed SEL during remote and hybrid learning. We describe what trauma-informed SEL looks like in practice and what considerations are advised as SEL continues in remote, hybrid, or in-person modalities. Our findings highlight the need for further training, collaboration, and cohesive approaches for the promotion of student SEL, and the importance of attending to both student and teacher well-being now and in the future.

**Author Contributions:** Conceptualization, R.S.L., R.J.L. and A.V.B.; methodology, R.S.L., R.J.L. and A.V.B.; formal analysis, R.S.L., R.J.L. and A.V.B.; writing—original draft preparation, R.S.L., R.J.L. and A.V.B.; writing—review and editing, R.S.L., R.J.L. and A.V.B.; supervision, A.V.B. All authors have read and agreed to the published version of the manuscript.

**Funding:** This research received no external funding.

**Institutional Review Board Statement:** The study was approved by the Institutional Review Board of the University of California, San Diego (protocol code #210376, approved 30 March 2021).

**Informed Consent Statement:** Informed consent was obtained from all subjects involved in the study.

**Data Availability Statement:** The data presented in this study are available on request from the corresponding author. The data are not publicly available to protect participants' confidentiality.

**Acknowledgments:** We would like to thank the teachers who participated in our study.

**Conflicts of Interest:** The authors declare no conflict of interest.

## Appendix A. Codebook

| Code | Definition | Excerpt | Frequency |
|---|---|---|---|
| Strategies: Focusing on Relationships | Building positive student–teacher relationships as well as peer friendships | "I just say hello or good morning or hi, especially for students that are entering the Zoom, because I want them to know that I know that they're there." (Morgan, interview) | 35 |
| Strategies: Building Routines and Predictability | Developing structures and norms in the classroom | "We need to do this [morning meetings] every day because otherwise there are just certain kids who, you could tell, would never speak or would never have their camera on." (Erin, interview) | 27 |
| Strategies: Creating Space to Identify and Share Feelings | Creating opportunities for students to recognize, identify, and share in a safe virtual space | "So I would ask them, it can be something simple, like I would get a lot of those 'How Are You Feeling' charts, like I would do one with emojis, or there's a lot of different ones on the Internet." (Erin, interview) | 27 |
| Strategies: Incorporating Movement, Mindfulness, and Play | Taking breaks from remote instruction to initiate physical activity, lighthearted fun, and mindfulness | "We did a lot of mindfulness breathing motions, we did a lot of Go Noodle." (Rachel, interview) | 34 |
| Strategies: Implementing Culturally Affirming Practices | Fostering inclusivity and encouraging students to celebrate their diverse backgrounds | "It [culturally affirming practice] really does help students to feel seen and to have even just little bits of their history understood." (Shelby, interview) | 8 |
| Strategies: Providing Student Choice and Leadership Opportunities | Providing opportunities for student agency in their schedule or learning; chances for students to take on responsibility and leadership | "We still managed to have classroom jobs." (Shelby, interview) | 18 |
| Strategies: Engaging and Collaborating with Families | Communicating with families to build relationships and join together to support students | "I connected deeply with their [the students'] families, because we had to communicate." (Elena, interview) | 21 |
| Challenges: Disconnection | Challenges to genuine interpersonal connection in a remote environment | "Can't see my students' faces!! Don't know if they are smiling and happy or frowning and sad." (Ashley, survey response) | 35 |
| Challenges: You Didn't Know Who was Listening to You | Barriers and concerns when others could listen in on classroom conversations | "You didn't know who was listening. You didn't know who was sitting next to that kid." (Natalie, interview) | 6 |
| Challenges: Teacher Burnout | Stress and exhaustion due to demands of virtual teaching amidst a global pandemic | "I would find myself just completely drained, out of energy, every single day." (Sonjia, interview) | 24 |

**Appendix B. Participant Demographics and Teaching Background (*n* = 26)**

| Participant | Gender | Race, Ethnicity | Years at Site | School Type | Student Grade Level | % of Students in Poverty |
|---|---|---|---|---|---|---|
| Alexis | Female | Black, non-Latino/a/x | 6–10 years | Public charter | Elementary | 75–100% |
| Ashley * | Female | White, non-Latino/a/x | 6–10 years | Public | Elementary | 75–100% |
| Anika * | Female | White, Asian, non-Latino/a/x | 6–10 years | Public | Elementary | Less than 60% |
| Clara | Female | White, Latino/a/x | 26–30 years | Public | Elementary | 75–100% |
| David * | Male | White, non-Latino/a/x | 17–25 years | Public | Elementary | 75–100% |
| Elena * | Female | White, non-Latino/a/x | 17–25 years | Public | Elementary | 60–74% |
| Erin * | Female | White, non-Latino/a/x | 6–10 years | Public | Elementary | Less than 60% |
| Frederick * | Male | White, Latino/a/x | 17–25 years | Alternative public | High | 75–100% |
| Gloria | Female | Other (race not provided), Latino/a/x | 3–5 years | Public | Elementary | Less than 60% |
| Julieta | Female | White, Latino/a/x | 3–5 years | Public | Middle | 75–100% |
| Kendall * | Female | Asian, non-Latino/a/x | 6–10 years | Public | Elementary | 75–100% |
| Lee | Preferred not to say | Preferred not to say, non-Latino/a/x | 17–25 years | Public | Elementary | 75–100% |
| Loren * | Non-binary/third gender | Asian, non-Latino/a/x | 6–10 years | Public charter | Middle | Less than 60% |
| Mara | Female | White, Latino/a/x | 3–5 years | Public | Elementary | 75–100% |
| Morgan * | Female | White, non-Latino/a/x | 3–5 years | Alternative private | Middle | 75–100% |
| Natalie * | Female | White, non-Latino/a/x | 26–30 years | Public | Elementary | 75–100% |
| Naomi | Female | Asian, non-Latino/a/x | 1–2 years | Alternative private | Middle/High school | Less than 60% |
| Priya * | Female | Asian, non-Latino/a/x | 3–5 years | Public | Elementary | 60–74% |
| Rachel * | Female | White, non-Latino/a/x | 17–25 years | Public | Elementary | 60–74% |
| Ramona | Female | White, Latino/a/x | 17–25 years | Alternative public | Elementary | 75–100% |
| Sandra * | Female | White, non-Latino/a/x | 1–2 years | Alternative public | High | 75–100% |
| Sarita * | Female | White, Latino/a/x | 6–10 years | Public | Elementary | 75–100% |
| Shelby * | Female | White, non-Latino/a/x | 11–16 years | Public charter | Elementary | Less than 60% |
| Sonjia * | Female | White, non-Latino/a/x | 6–10 years | Public | Elementary | Less than 60% |
| Victoria | Female | Other (race not provided), Latino/a/x | 6–10 years | Public | Elementary | 75–100% |
| Wendy | Female | Asian, non-Latino/a/x | 11–16 years | Public | Elementary | 75–100% |

* Interviewee.

## Appendix C. Participant Descriptive Statistics by Data Collection Method

| | Surveyed Participants Only (*n* = 10) | Surveyed and Interviewed Participants (*n* = 26) |
|---|---|---|
| **Gender** | | |
| Female | *n* = 9 (90.0%) | *n* = 23 (84.6%) |
| Male | *n* = 0 (0.0%) | *n* = 2 (7.7%) |
| Non-binary/third gender | *n* = 0 (0.0%) | *n* = 1 (3.8%) |
| Preferred not to say | *n* = 1 (10.0%) | *n* = 1 (3.8%) |
| **Race** | | |
| White | *n* = 4 (40.0%) | *n* = 17 (65.3%) |
| Asian | *n* = 2 (20.0%) | *n* = 5 (19.2%) |
| Preferred not to say | *n* = 1 (10.0%) | *n* = 1 (3.8%) |
| Black | *n* = 1 (10.0%) | *n* = 1 (3.8%) |
| Other | *n* = 2 (20.0%) | *n* = 2 (7.7%) |
| **Ethnicity** | | |
| Latino/a/x | *n* = 6 (60.0%) | *n* = 8 (30.8%) |
| Non-Latino/a/x | *n* = 2 (20.0%) | *n* = 18 (69.2%) |
| **Years at site** | | |
| 1–2 years | *n* = 1 (10.0%) | *n* = 2 (7.7%) |
| 3–5 years | *n* = 3 (30.0%) | *n* = 5 (19.2%) |
| 6–10 years | *n* = 2 (20.0%) | *n* = 9 (34.6%) |
| 11–16 years | *n* = 1 (10.0%) | *n* = 2 (7.7%) |
| 17–25 years | *n* = 2 (20.0%) | *n* = 6 (23.1%) |
| 26–30 years | *n* = 1 (10.0%) | *n* = 2 (7.7%) |
| **School type** | | |
| Public | *n* = 7 (70.0%) | *n* = 18 (69.2%) |
| Public charter | *n* = 1 (10.0%) | *n* = 3 (11.5%) |
| Alternative private | *n* = 1 (10.0%) | *n* = 2 (7.7%) |
| Alternative public | *n* = 1 (10.0%) | *n* = 3 (11.5%) |
| **Student grade level** | | |
| Elementary | *n* = 8 (80.0%) | *n* = 20 (76.9%) |
| Middle | *n* = 1 (10.0%) | *n* = 3 (11.5%) |
| Middle/High school | *n* = 1 (10.0%) | *n* = 1 (3.8%) |
| High school | *n* = 0 (0.0%) | *n* = 2 (7.7%) |
| **% of students living in poverty** | | |
| Less than 60% | *n* = 2 (20.0%) | *n* = 7 (26.9%) |
| 60–74% | *n* = 0 (0.0%) | *n* = 3 (11.5%) |
| 75–100% | *n* = 8 (80.0%) | *n* = 16 (61.5%) |

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
