# Peer review of "Social and Emotional Learning during Pandemic-Related Remote and Hybrid Instruction: Teacher Strategies in Response to Trauma"

_education, doi:10.3390/educsci13040411_

Round 1

Reviewer 1 Report

This article reports on a primarily qualitative study conducted in May/June 2021 exploring the strategies teachers used and the challenges teachers faced in delivering SEL to students during the COVID pandemic when teaching remotely or in hybrid formats. The authors consider SEL from a trauma-informed practices perspective. The article is clearly written and effectively gathered insightful information about teachers’ perceptions of teaching SEL in remote/hybrid format.

              Though the article findings are interesting, and the consideration of trauma-informed practices alongside SEL is appropriate, the main contribution of the article is unclear and the description of methodology could be strengthened. Though the authors clearly describe how the strategies they delineate in the results align with trauma-informed practice recommendations, it is unclear how these results add anything new. The authors might consider how their results could expand or provide nuance to the recommendations of trauma informed practices. In the discussion of challenges, the authors refer to others’ findings rather than their own findings. This seems more appropriate for the conclusion. The authors might explore how these challenges, though no longer relevant since most teaching is now in-person again, might influence teacher’s future experience. What does it mean that teachers felt disconnected, does this result point to possible subsequent challenges that teachers must now deal with? What about those parents that were listening in, does this result suggest possible challenges and benefits going forward? Teacher burnout has been an ongoing issue in education, what are the implications of these teachers’ experience of that burnout being exacerbated. This seems like a very important finding (one that others have also written about). The authors might find a way to link the challenge paragraph (lines 619-627) to this issue of teacher burnout to make it more clear to the reader how this content relates to their results. In general, I suggest the authors consider and write about the implications of their results for these teachers, and other teachers like these, going forward. This experience of hybrid/remote teaching and the COVID-19 pandemic has had an impact on teachers and their students that will have lasting effects, since trauma like this is not momentary. The authors revisit their results to consider how what they learned might inform trauma-informed teaching practices in delivering SEL to students who are now dealing with the after- and ongoing-effects of this traumatic time. With this approach to the discussion, the authors may have a better basis for providing recommendations in their conclusions; including recommendations for future research on how teachers are dealing with the fallout from this remote/hybrid instructional time.

              Considering the primary qualitative nature of the study and the subjective learning the authors refer to that informed the study (lines 206-207), this article should include a brief researcher positionality section describing the research team (referenced on line 220) to increase the rigor of the study. Similarly, the authors should describe in more detail how they worked together to “interrogate the themes” to ensure the validity of their findings (line 233). Related to the quantitative data, it is unclear what they contribute, other than describing the participants. The authors should consider this as they revise. As I see it, the authors could use the quantitative data to supplement their qualitative data and/or use the quantitative data solely describe their sample (take into account next comment also). It would also be helpful to know if there are any differences (statistical if possible) between those who did not complete the survey (if the authors know the approximate demographic make-up of the list-serv members) and between those who completed the survey and those who were subsequently interviewed. This would help the reader better know who was in the sample. The authors might consider adding a table with descriptive statistics of these three (or two if the list serve demographics are unknown) groups, rather than (or in addition to) the aggregated (section 2.5) and individual (Appendix B) participant descriptive information.

Specific comments:

·       The sentence on lines 29-30 is awkward, consider revising for clarity

·       Remove ‘has’ at end of line 31

·       Check the accuracy of the sentence “These findings have been evident across, age, socioeconomic status, and ethnicity and culture (Durlak et al., 2011; Zins & Elais, 2007)” on line 63-64)  A recent systematic review of SEL programs by Cipriano et al. (2023 in Review of Educational Research), indicates otherwise. Revise accordingly.

·       Briefly describe/define how you did attribute coding (line 224)

·       Typo (and) on line 534

·       Better describe/address limitation of bias (see previous comments) on lines 588-590 and make sure to include the difference between your survey and interview sample, since most of your results appear to be based on the interview sample (if this is not the case, then you may want to describe/highlight your qualitative survey data more specifically.

·       The sentence on lines 626-628 is awkward, revise for clarity.

·       Expand on the statement “which can be especially important when supporting students who are experiencing trauma” on line 637. The rationale for this statement is not obvious and needs to be provided for the reader to make sense of it. Also, consider how this relates to supporting students who experienced trauma – since one of the challenges of supporting students who have experienced trauma is because the effects of trauma don’t just go away when the experience is behind them.

·       I’m not sure how the findings you describe ‘highlight the need for whole-school training and approach so that all staff can support one another in promoting student SEL “ on lines 665-667. Either clarify the connection to your results or consider points that relate more closely to your results to include in your conclusions.

Reviewer 2 Report

Describe Responsive Classroom. What are the features of the system/program. How is Responsive Classroom connected to the study? Author (s) could also describe the components of morning meeting. Was it a key practice for all educators involved in the study?

Reviewer 3 Report

This paper describes a survey and interview process with a small group of teachers in California following remote and hybrid teaching during the COVID pandemic.  It is quite well written and provides a good overview of the rationale for and tenets of SEL and trauma-informed practices. My only concern is that it is limited in the amount of new information or knowledge that it provides.  The strategies and challenges covered feel like a rehash of most of what I have read about education in the wake of COVID.  The portion that I found most compelling was the discussion and recommendations section pertaining to the lack of orchestrated approaches with vetted materials. I would love to see a deeper reflection in this area.  The authors briefly note the inefficiency of requiring teachers (who are already burned out and overwhelmed) to search for their own resources and gauge which might be most effective. They might elaborate on ways that school systems do or could manage this in a more structured way, calling upon those with specific training (social workers, counselors) to lead the work.  This also gives room to advocate for a better ratio of students to counselors!  They could also bring in some of the concerns with the current approach. For instance, mindfulness activities have the potential to trigger panic attacks in a subset of students, requiring teachers to monitor for this possibility and have resources available just in case, but many aren’t aware of this. 

Overall, it seems a lengthy paper for the level of new information that it adds to the field. However, if this information were summarized more succinctly with deeper discussion of solutions, it could be a useful addition to the literature. 
